# ANNOTATION BOOTSTRAPPING: REINFORCING VISUAL PRE-TRAINING USING UNLABELLED IMAGES

## ABSTRACT

A common approach to learning from unlabeled images is to train models to satisfy invariances on these images, such as consistency under augmentations or crops. Despite successes on Imagenet, these approaches struggle to learn from larger uncurated datasets like web crawls or video, where such inductive biases only weakly hold. How can we more effectively learn from broader datasets? Instead of training models to be invariant across views, we study an alternative approach encouraging model representations to be *predictive* of important semantics of adjacent views of an image. We concurrently train a model to predict semantic annotations from images (generated either self-supervised, or from auxiliary datasets); and bootstrap the model's semantics by predicting, given a cropped view of an image and the coordinates for a nearby crop, the model's annotation distribution for the neighboring view. A core strength of this approach is the ability to extract information universally from both unlabelled and labelled image data, incorporating captions, bounding boxes, and other annotations when they are present. Our experiments show that annotation propagation improves pre-training on unlabelled datasets in the wild, including video datasets like EpicKitchens, scene datasets like COCO, and uncurated web-scale image datasets like CC12M.

## 1 INTRODUCTION

The ability to learn from large unlabelled datasets has served as the impetus for much of the recent progress in deep learning. In natural language processing, text corpora fed into generic unsupervised objectives like next-token or masked prediction have enabled increasingly capable NLP models. Even though unlabelled visual data is similarly plentiful and easy to collect — on the internet, from video, from embodied agents, and beyond — learning from such images has proven a greater challenge, in part because they are a raw signal with redundancy, low information density and noise.

To learn useful semantics without labels, a common approach is to train models to respect certain invariances on images, such as consistency of representations between views of an image transformed by augmentations, crops, or masking. By representing two different crops of an image with similar features, models may learn high-level features insofar as guessing which crops go together requires recognizing semantically salient objects (e.g., that a lion should be associated with the savannah). Despite their empirical successes, these consistency objectives introduce inductive biases tailored to pre-training datasets like Imagenet and downstream metrics like object classification. It is unclear how to generalize these inductive biases to a broader set of images, like web crawl or video data, and towards other downstream tasks like object detection or embodied action recognition.

In this paper, we consider a framework in which a model attempts to learn semantic relationships between images and "annotations", and then uses its learned model to predict these semantic annotations across neighboring views of images. We call this framework annotation bootstrapping, since the core of the pre-training process involves using a model's predictions about semantic annotations from one view to bootstrap the predicted semantics of other neighboring views of an image. The annotation and bootstrapping objectives are decoupled, which means that we may flexibly steer pre-training by curating or changing the annotation data, while still using the entire unlabelled image corpus to "bootstrap" these semantics across nearby images. Unlike invariance-based approaches to learning from unlabelled data, where learned features are defined opaquely by some interplay

between augmentations or objectives, our approach offers a controllable lever to guide the features learned through pre-training.

We investigate our pre-training scheme on unlabelled data distributions where self-supervised approaches typically degrade, like web-scraped datasets used for VLM training, video datasets, and datasets that are not object-centric. In the completely self-supervised setting, we synthesize annotations by sampling two transformed views of an image, we find that our joint objective leads to better representations than those that train for consistency between views, or otherwise directly predict in pixel space. Since the annotation and propagation processes are completely decoupled in our framework, we find that we can also improve performance by, for example, learning relationships between crops on Imagenet (where the inductive bias of consistency fits well), and then propagating these annotations on video or scene datasets, where it does not. We find also that our framework is amenable to leveraging annotations from other sources. For example, we show that using web-scraped captions as annotation targets (effectively turning the problem into one of predicting distributions over targets) allows us to learn vision-language models more capable than those learned by CLIP, and other approaches that combine CLIP with self-supervised objectives.

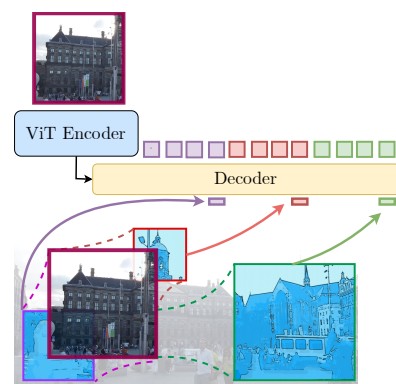

Figure 1: In annotation bootstrapping, we learn from unlabelled images by training a model to predict semantic annotations associated with one crop of an image from other crops of the image. The key idea in our framework is that we may learn to solve this prediction task only using unlabeled data by *bootstrapping* off the model's own predictions.

The primary contribution of this paper is annotation propagation, an approach for pre-training on unlabelled image data that jointly learns image-annotation relationships and predicts distributions of these annotations across image views. Compared to other self-supervised methods, this approach can more readily ingest uncurated image data without sacrificing image performance and allows us to easily incorporate any (weakly-) supervised data into the pre-training process.

## 2 RELATED WORK

**Self-supervised learning.** Self-supervised methods learn from unlabelled image data generally in two ways: by generation or by enforcing representational consistency. Generative approaches predict raw pixels (He et al., 2021) or other low-level features (Xie et al., 2021; Bao et al., 2021) from a corrupted or masked image input; the quality of learned image semantics can depend significantly on the pixel statistics of pre-training images (cite). Consistency-based approaches instead train representations to be similar across image transformations like augmentations or masks. Consistency can be enforced directly via instance-level discrimination (van den Oord et al., 2018; Chen et al., 2020a) or implicitly in the limit of self-distillation (Grill et al., 2020; Caron et al., 2019; He et al., 2019; Chen & He, 2020; Assran et al., 2023). These methods also mimic qualities of the pre-training distribution (cite uniform prior), and degrade on atypical images and uncurated datasets. One challenge with self-distillation in latent space admits pathological solutions (like the trivial representation $\phi = 0$), often requiring advanced techniques such as logit sharpening (Caron et al., 2021; Oquab et al., 2023) or asymmetrical predictors (Grill et al., 2020).

**Vision-language pre-training.** On vision-language datasets, prior work has found gains in combining weakly supervised losses (Radford et al., 2021; Jia et al., 2021; Zhai et al., 2023) with self-supervised ones: Mu et al. (2021) combines CLIP with a SimCLR objective using an auxiliary head, Li et al. (2021) jointly runs CLIP and SimCLR both on the same representation, Naeem et al. (2023) combines SigLIP with a DINO objective. These methods improve the data efficiency of contrastive vision-language training and improves performance for more fine-grained tasks like segmentation and prediction; however, other evidence however suggests that simpler solutions also offer the same

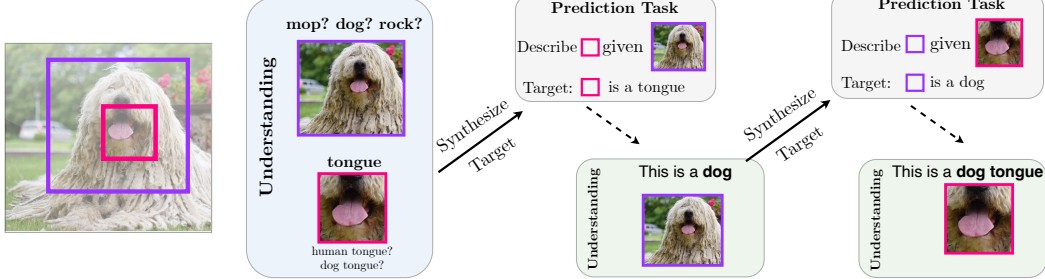

Figure 2: **Conceptual illustration.** A toy model illustrating how bootstrapping annotations can help models propagate visual semantic understanding while pre-training on unlabeled data. By asking the model to predict from the purple crop, the model's outputs on the pink crop, and vice versa, the model can learn features that better explain both crops of a scene. More generally, training a model to predict details about the neighboring scene encourages it to acquire a better semantic understanding of its own view, thereby improving the data used to train the model in the future.

invariances, for example by increasing augmentation in the CLIP objective (Fini et al., 2023) or by scaling data (Weers et al., 2023).

**Semi-supervised learning**   While our paper focuses on unlabelled image pre-training guided by descriptive annotations like free-form text, it is adjacent to a longer line of semi-supervised approaches learning with partially annotated class labels. Two techniques are common: combining a supervised classification loss with an self-supervised objective on unlabelled data (Pathak et al., 2016; Chen et al., 2020b; Zhai et al., 2019a; Xie et al., 2019), and using the supervised dataset to create pseudo-labels (Lee et al., 2013) for unlabelled images (Xie et al., 2019; Pham et al., 2020). While both pseudo-labeling and our bootstrapping generate target predictions using the model's outputs, we note one important difference: pseudolabelling creates labels for a different model to train, while our approach is designed to enable a model to learn from *its own* synthetic targets.

## 3   PRE-TRAINING BY BOOTSTRAPPING ANNOTATIONS

We propose the annotation bootstrapping framework, which uses unlabelled image data to propagate the model's understanding of image semantics across "neighboring" views of an image. The core principle is the bootstrap: that the current model's predictions about a given image can be used to create a prediction target for *itself* – just for a different image input.

In this framework, models learn in two threads; one that predicts semantic annotations from images, and one that "bootstraps" these annotations across different views of an image to teach models how to predict the semantics of its surroundings. Intuitively, the bootstrapped update is simple: we take an crop of an unlabelled image and transform it (e.g. by zooming in, panning left, etc); the model is trained to predict, given the first image and a description of the transform applied, the annotations detected by the model in the transformed image. As we illustrate conceptually in Figure 2, these two threads are synergistic; as the model improves its ability to predict semantics about the scene around it, it acquires a better semantic understanding of its own scene, thereby improving the targets used to train the model in the future.

**Learning to predict annotations from images.**   The first step to formalizing our framework is to specify what it means to model (and eventually predict across views) useful semantic concepts. We co-opt the terminology of image annotations, arbitrary descriptors of useful semantic concepts in images. A useful mental model is to imagine annotations as text descriptions of an image, although they can be masks, captions, audio, or even other images. The learning problem is to use a dataset of image-annotation pairs $\mathcal{D}_a = \{(x_i, \ell_i)\}$ to predict the annotation distribution $p(x, \ell)$; when modeled contrastively, yields a well-known learning problem (Poole et al., 2019; Eysenbach et al., 2024).

**Note 1** *The following optimization yields the annotation distribution up to an additive constant:*

$$\log \frac{p(\ell|x)}{p(\ell)} + C = \arg\max \mathbb{E}_{(x_i,\ell_i)_{i=1}^n \sim p}\Big[\sum_i \log \frac{\exp(f(x_i,\ell_i))}{\sum_j \exp(f(x_i,\ell_j))} + \log \frac{\exp(f(x_i,\ell_i))}{\sum_j \exp(f(x_j,\ell_i))}\Big] \quad (1)$$

We choose this framework because it encompasses many common learning algorithms, supervised, weakly-supervised, or self-supervised, depending on the choice of annotations. When annotations are textual captions, this corresponds to the CLIP objective (Radford et al., 2021); when annotations are defined to be augmented crops of the same image $\ell = \text{augment}(\text{randomcrop}(x))$, this becomes the SimCLR objective, and when annotations are discrete class labels, this corresponds to a (label-smoothed) classification objective. In our experiments, we will consider annotations of both textual types (corresponding to a base CLIP loss), and images (corresponding to a base SimCLR loss) to show how our method can be applied in both fully self-supervised AND weakly supervised settings.

**Learning on unlabelled data by bootstrapping annotations.** For a model that is being trained to predict semantic annotations associated with an image, we can pre-train on unlabelled data to "bootstrap" these concepts to a broader set of images. Intuitively, the idea is that, by taking two crops of an unlabelled image, we can use the model's outputs to generate target responses for questions of the form "What will be the annotations we see if we zoom out from the current view? or "what semantic concepts exist to the right of the current view?".

Formally, we model image transformations as a dynamical system that applies some transformation $a$ to an image $x$ to realize a new image $x'$ from some distribution $p(x'|x,a)$. The bootstrapping objective is to predict the annotation distribution that is associated with the transformed image

$$p(\ell|x,a) = \mathbb{E}_{x' \sim p(x'|x,a)}[p(\ell|x')]. \quad (2)$$

In words, the learning problem is – given an image $x$ and a description of an image transformation $a$ – to predict the annotation distribution for the transformed image $x'$. This captures a wide range of image transformations – whether transforming an image self-supervised (e.g. zooming, panning, rotating, cropping) as we study in this paper, but also includes other types of image transformations such as "the image resulting when I step one frame forward in a video".

We will learn to estimate this distribution by bootstrapping our predictions about transformed images using our current model's predictions about annotations associated with the current image:

**Note 2 (Bootstrapping objective)** *To estimate the distribution of annotations ensuing from applying a transformation $a$ to an image $x$, $g_\theta(x,a,\ell)$, we will bootstrap using our model's predictions $f_\theta(x',\ell)$ from the base contrastive prediction task:*

$$\arg\max_g \mathbb{E}_{\{\ell_i\}_{i=1}^n \sim p(\ell),(x,a,x') \sim p}\left[\sum_i \frac{\exp(f_\theta(x',\ell_i))}{\sum_j \exp(f_\theta(x',\ell_j))} \log \frac{\exp(g_\theta(x,a,\ell_i))}{\sum_j \exp(g_\theta(x,a,\ell_j))}\right]. \quad (3)$$

*Fully optimizing the base contrastive loss and bootstrapping loss recovers the desired distribution.*

In form, this objective looks like model distillation, but has an important crucial difference: the prediction and target are not being done on the same image. While distillation must transfer knowledge from one model to another, this approach instead transfers knowledge *from the model to itself*, by presenting the information in the context of a different image. Second, this objective does not require us to have paired access to annotations, only samples from the marginal distribution. This is the crucial step that allows us to leverage unlabelled image data: to optimize this objective, we only need to be able to sample images, and to be able to transform these images as we desire. In the next section, we describe a concrete instantiation of this algorithm, which will learn to propagate annotation distributions over a very general class of bounding box transformations.

## 4 PRACTICAL IMPLEMENTATION OF ANNOTATION BOOTSTRAPPING

We now describe a practical implementation of annotation bootstrapping (abbreviated AB for short). This implementation can be wrapped around any base annotation prediction task, but we will describe two corresponding to settings of increase. $AB_{\text{SimCLR}}$ uses SimCLR (Chen et al., 2020a) as

---

**Algorithm 1** Annotation Bootstrapping (AB)

---

▷ **Base contrastive annotation update**

Let $f_\theta(x, \ell) = \phi(x)^\top \psi(\ell)$ be the contrastive logits for the image $x$ and annotation $\ell$
Sample annotation batch $\mathcal{B}_a = \{(x_i, \ell_i)\}_{i=1}^{n_a} \sim \mathcal{D}_a$
Compute contrastive loss on annotation batch:

$$\mathcal{L}_{annotation} = \sum_i \log \frac{\exp(f(x_i, \ell_i))}{\sum_j \exp(f(x_i, \ell_j))} + \log \frac{\exp(f(x_i, \ell_i))}{\sum_j \exp(f(x_j, \ell_i))}$$

▷ **Annotation bootstrapping update**

Let $g_\theta(x, a, \ell) = \bar{\phi}(x, a)^\top \bar{\psi}(\ell)$ be the logits for the image $x$, transformation $a$, and annotation $\ell$
Sample unlabelled images $\mathcal{B}_u = \{x_i^u\}_{i=1}^n \sim \mathcal{D}_u$ and copy $\{\ell_i\}$ from the annotation batch
**for** each image $x$ in the unlabelled image batch **do**
    Sample and crop the images to a source bounding box $\mathbf{bb}_s$ and target bounding box $\mathbf{bb}_t$
    Optimize Equation 3 with $x = \text{crop}(x, \mathbf{bb}_s)$, $x' = \text{crop}(x, \mathbf{bb}_t)$, $a = \mathbf{bb}_{s \to t}$)

$$\mathcal{L}_{propagation} = \sum_i \frac{\exp(f_{\theta_{ema}}(x', \ell_i))}{\sum_j \exp(f_{\theta_{ema}}(x', \ell_j))} \log \frac{\exp(g_\theta(x, a, \ell_i))}{\sum_j \exp(g_\theta(x, a, \ell_j))}. \tag{4}$$

**end for**

---

the base loss to learn in the fully self-supervised setting (where annotations are images generated by augmentations and crops); $AB_{CLIP}$ learns using CLIP (Radford et al., 2021) in the weakly-labelled setting (where annotations are textual strings like image captions). The methods are nearly identical in both cases, with slight implementation differences to handle text and image annotations respectively.

**Transformation Distribution:** To instantiate the annotation bootstrapping framework, we must specify a set of image transformations that we are interested in modelling. Rather than manually specifying a small set of common transformations (like zooming, panning, etc), we we will consider any transformation that corresponds to "changing the bounding box view" of the image. More specifically, when we generate two views from an image $I$ by cropping the image with different bounding boxes: $x = \text{crop}(I, \mathbf{bb}_s)$ and $x' = \text{crop}(I, \mathbf{bb}_t)$, we will say that the image transformation $\mathbf{bb}_{s \to t}$ is the process that transforms the view $x$ to the view $x'$.

**Model:** We parameterize the contrastive distribution as an inner product between an image representation $\phi(x)$ and an annotation distribution $\psi(\ell)$; similarly for the prediction head, an inner product between the image-action representation $\bar{\phi}(x, a)$ and $\bar{\psi}(\ell)$. In $AB_{SimCLR}$, as in Chen et al. (2020a), the annotation representation is identical to the image head; in $AB_{CLIP}$, we use a standard CLIP text transformer from Radford et al. (2021) to parameterize the annotation head.

We use the same network for the image and image-action representation: an encoder-decoder Transformer, using a Vision Transformer (Dosovitskiy et al., 2020) as the image encoder, and a standard Transformer decoder architecture that attends to the ViT embeddings via cross-attention. This architecture is a standard recipe (Tschannen et al., 2024) and allows us to flexibly parameterize action transformations. Recalling that image transformations corresponding to changing the bounding box view, we represent it as a set of four tokens, each corresponding to the four corners of the new bounding box within the relative coordinate frame of the current. We experimented with both discrete learned embeddings and fixed positional Fourier embeddings, and found no substantive different in performance. For the image representation $\phi(x)$, we pass in the "identity action" (corresponding to keeping the bounding box exactly the same).

**Annotation Loss:** Optimizing the attention loss is basically exactly the same as in CLIP / SimCLR; we sample a batch of images and annotations $\mathcal{B}_a = \{(x_i, \ell_i)\}_{i=1}^n$, and optimize the contrastive loss:

$$\mathcal{L}_{annotation} = \sum_i \log \frac{\exp(f(x_i, \ell_i))}{\sum_j \exp(f(x_i, \ell_j))} + \log \frac{\exp(f(x_i, \ell_i))}{\sum_j \exp(f(x_j, \ell_i))} \tag{5}$$

Learning Annotations · Propagating Annotations

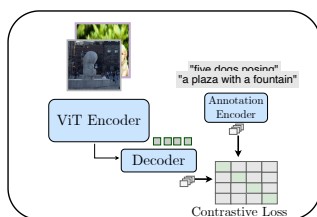
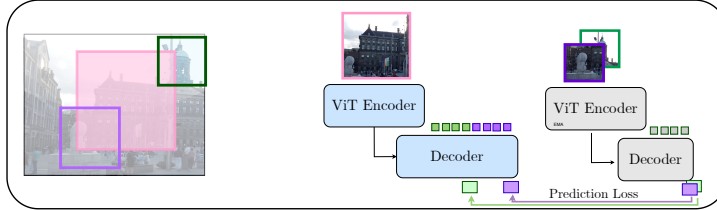

**Bootstrapping Loss:** We will sample a batch of images $\mathcal{B}_u = \{x_i\}_{i=1}^{n_u}$, and copy (only) the annotations from the labelled batch $\{\ell_i\}_{i=1}^{n_\ell}$. For each image, we will create $M$ views of the image by sampling $M$ random bounding boxes using Inception-style random cropping (in torch, using randomresizedcrop) to create the set of images. Note that we do not use any color augmentations or random flips. We find that adding these makes the prediction task harder and decreases the final quality of learned representations, as we see in our ablations. We then use a lagging EMA network to obtain the predicted annotation distributions at the current stage, and optimize the objective in Equation 3 with all $M^2$ pairs of views, $(x = x_i, x' = x_j, a = \mathbf{bb}_{i \to j})$.

In practice, we found no substantive difference in training efficiency in terms of number of views seen, but faster wall-clock speeds with multi-view training. We follow the recipe from DINO (Caron et al., 2021) to update the lagging EMA parameters, which uses an exponential moving average with a rate $\tau$ that is decayed from $0.004 \to 0$ via a cosine schedule.

## 5 EXPERIMENTS

In our empirical study, we measure the effectiveness of annotation propagation for learning useful features from unlabelled image datasets found "in-the-wild". We study both the fully unsupervised setting (where propagation occurs in the space of image-image relationships, based on the SimCLR loss), and in the weakly labelled setting (where propagation occurs in the space of image-caption relationships, based on the CLIP objective). Our experiments seek to answer three main questions:

1. How does our approach compare to invariance-based and pixel-predictive approaches?
2. Can we use semantic annotations from one dataset to learn on a different unlabelled dataset?
3. What factors in the prediction objective affect the quality of learned representations?

### 5.1 SETUP

**Datasets.** We evaluate on four datasets representative of the many types of unlabelled images typically available: Imagenet (Russakovsky et al., 2014), a well-curated, balanced, and image-centric benchmark heavily used by prior work; CC12M (Changpinyo et al., 2021), a dataset of captioned images used for vision-language pre-training that is relatively uncurated and contains a wider range of concepts than Imagenet; COCO (Lin et al., 2014) a dataset of scenes each containing many (potentially small) objects, and Epic-Kitchens (Damen et al., 2020), a video dataset containing many real-world scenes in homes. Note that CC12M is a dataset of links, so links deteriorate due to rot and redirects; the version we collected (Beaumont, 2021) has 8.7 million images.

**Training.** We standardize training by running all methods on all datasets using ViT-S vision encoders (and S-sized text encoders in the weakly labeled setting) for $800M$ seen images (each view is counted separately). For ImageNet, this corresponds to approximately $620$ epochs. All models are trained with AdamW, weight decay, gradient clipping, and using a cosine decay schedule – specific hyperparameters are taken from respective papers when they are provided. In the fully self-supervised setting, we compare our approach to SimCLR (Chen et al., 2020a), DINO (Caron et al., 2021), and MAE (He et al., 2021); we emphasize that our experimental goal is not to claim state-of-the-art performance on unsuperivsed benchmarks, but rather to evaluate the merits and deficiencies of *prediction* in "annotation" space, and how it relates to more common design patterns like consistency and generative pixel prediction. In the weakly-supervised setting, we compare our approach

Table 1: Downstream classification metrics when pre-training fully unlabelled on ImageNet and CC12M, and when using weakly labelled supervision on CC12M. **\*Avg. Cls** averages the classification accuracy over the four benchmarks in Beyer et al. (2023): Food101, Oxford IIIT Pets, Resics45, and Sun397.

| PRETRAIN DATASET | METHOD | IMAGENET | AVG CLS* | CLEVR/DEPTH | CLEVR/COUNT |
|---|---|---|---|---|---|
| ImageNet (No Labels) | SimCLR | 70.0 | 80.1 | 76.9 | 86.0 |
| | DINO | 72.2 | 82.8 | 80.0 | 88.1 |
| | MAE | 65.0 | 77.7 | **81.7** | 88.6 |
| | I-JEPA | 64.5 | 79.0 | 81.0 | 88.8 |
| | AB$_{\text{SimCLR}}$ | **73.6** | **83.7** | 81.4 | **89.3** |
| CC12M (No labels) | SimCLR | 64.9 | 74.3 | 78.3 | 87 |
| | DINO | 67.8 | 79.5 | 79.5 | 87.1 |
| | MAE | 60.1 | 74.5 | **81.4** | 88.5 |
| | I-JEPA | 60.0 | 76.0 | 80.1 | **90.2** |
| | AB$_{\text{SimCLR}}$ | **68.7** | **80.1** | 82.1 | 89.4 |
| CC12M (w/ Captions) | CLIP | 70.0 | 82.4 | 73.1 | 84 |
| | SLIP +SimCLR (Mu et al., 2021) | 69.0 | 81.1 | **77.3** | 88.7 |
| | SiLC +DINO (Naeem et al., 2023) | 71.0 | 83.6 | 73.9 | 86.6 |
| | AB$_{\text{CLIP}}$ | **74.6** | **84.0** | 78.0 | 92.9 |

to CLIP (Radford et al., 2021), SLIP (Mu et al., 2021), which combines CLIP with a SimCLR loss, and SILC (Naeem et al., 2023), which combines CLIP with a DINO loss.

**Evaluation.** To avoid overfitting to benchmark numbers like Imagenet, we evaluate on a wider set of benchmarks, building on the probing strategy introduced by Beyer et al. (2023), which probes performance using a lightweight decoder that cross-attends with the ViT embeddings. This solution allows us a unified interface to evaluate any downstream task that can be cast as a sequential modeling problem (including classification, object detection, small object classification, captioning, etc). For Imagenet and CC12M, both which are relatively object-centric, we evaluate on downstream ImageNet, the classification tasks from Beyer et al. (2023), and the Clevr (Johnson et al., 2017) counting and distance prediction from (Zhai et al., 2019b) Images from EpicKitchens and COCO differ from these object-centric evaluation tasks; we evaluate on tasks naturally defined in their space (action recognition, object classification in video, and object detection). For full details about the evaluation setup, please see Appendix A.1

## 5.2 RESULTS

**Fully unlabelled pre-training.** We first evaluate annotation propagation in the fully unlabeled setting (AB$_{\text{SimCLR}}$), where our approach learns by learning to match two augmented views of an image together using a base SimCLR loss, and propagating annotations in the induced space of image-image relationships. We make two overarching observations in our experiments. First, on datasets and benchmarks where "invariance to crops and augmentations" holds , annotation propagation is synergistic to the base SimCLR loss, improving performance on average. On datasets and benchmarks where this inductive bias does not hold, SimCLR performance significantly degrades, but annotation propagation does not, indicating that the propagation objective also enables the model to learn a wider set of features useful for tasks beyond the base distribution.

On both Imagenet and CC12M (Table A.2, top), annotation propagation learns representations that outperform SimCLR, DINO, and MAE for downstream classification tasks (noting that for all methods and many downstream tasks, there is a uniform drop-off on CC12M, the less curated dataset). Our approach improves over SimCLR in downstream I1K performance by ∼4% on both datasets. We notice that SimCLR and DINO degrade relative to MAE on the Clevr tasks, while annotation propagation does not, indicating that our approach can avoid common failure modes associated with invariance-based self-supervised learning. The invariance to crops imposed by SimCLR and DINO is a particularly bad match for EpicKitchens and COCO (Table 2). On these domains, we find that annotation propagation outperforms these invariance-based approaches, but is equal or slightly worse than MAE across the board. While annotation propagation can enable the model to learn features beyond the invariances, it is not a cureall for a poor base loss.

Table 2: Downstream metrics for unlabelled pre-training on EpicKitchens and COCO.

| PRETRAIN DATASET | METHOD | IMAGENET | EK ACTION RECOGNITION | EK OBJECT DETECTION | EK OBJECT CLASSIFICATION |
|---|---|---|---|---|---|
| EpicKitchens | SimCLR | 48.1 | 20.3 | 0.299 | 70.4 |
| | DINO | 43.4 | 18.9 | 0.295 | 39.6 |
| | MAE | **43.5** | 20.8 | **0.387** | **76.5** |
| | I-JEPA | 38.7 | 18.5 | 80.1 | 21.5 |
| | AB$_{SimCLR}$ | 47.1 | 19.8 | 0.328 | 72.6 |

| PRETRAIN DATASET | METHOD | IMAGENET | COCO OBJECT DETECTION | COCO OBJECT CLASSIFICATION |
|---|---|---|---|---|
| COCO | SimCLR | 56.2 | 0.24 | 70.4 |
| | DINO | 56.1 | 0.24 | 70.5 |
| | MAE | **62.3** | **0.31** | **76.5** |
| | I-JEPA | 43.0 | 0.21 | 62.5 |
| | AB$_{SimCLR}$ | 59.7 | 0.27 | 72.6 |

**Learning with captions.** We next turn to evaluating annotation propagation in the weakly labelled setting, when the annotations are tokenized strings of text. Recall that in this setting, our approach learns by modelling text from images using a base CLIP loss, and propagates image-text relationships by modelling the distribution of captions closely associated with different crops of the image.

On CC12M (Table A.2, bottom), we see that weakly supervised methods across the board outperform their unsupervised equivalents; this matches empirical evidence that contrastive language-text methods are more capable of training on lower-quality image data. As discussed by Naeem et al. (2023), we find combining CLIP with a self-supervised objective, whether DINO or SimCLR, improves fine-grained reasoning on the ClevR benchmark tasks, but only marginal improvement on downstream classification tasks. In contrast, we see that annotation propagation obtains much stronger performance relative to these other approaches on most of the downstream metrics we evaluated, in particular improving by $4.6\%$ on downstream ImageNet probing performance over the base CLIP representations. We hypothesize that a significant component of this is that annotation propagation learns by making predictions about text distributions associated with other crops of an image, whereas the "unsupervised" representations optimized to be consistent in SLIP and SILC are only indirectly related to the CLIP representation (insofar that they use a shared backbone).

Thus far, we have only considered training in settings where the annotation and unlabelled data distributions are the same. However, recall that one of the conceptual strengths of annotation propagation is that the contrastive annotation loss and the predictive propagation loss are entirely decoupled, and do not need to be optimized on the same dataset. We now investigate how our approach can use an annotated dataset from an auxiliary source to improve the quality of unsupervised pre-training.

We begin by comparing the performance of different weakly-supervised methods for pre-training on COCO in Table 3, a dataset where we found invariance-based methods to struggle.

Table 3: Weakly supervised training on COCO

| TYPE OF ANNOTATIONS | METHOD | OBJECT CLS | DETEC -TION |
|---|---|---|---|
| COCO Captions | CLIP | 25.4 | 71.9 |
| | SLIP | 25.8 | 75.1 |
| | SILC | 21.8 | 71.2 |
| | AB$_{CLIP}$ | **29.7** | **76.7** |
| Bounding Boxes | CLIP | 31.6 | 76.4 |
| | SLIP | 28.3 | 76.3 |
| | SILC | 29.2 | 76.4 |
| | AB$_{CLIP}$ | **34.9** | **82.5** |

We use two auxiliary datasets sourced directly from COCO: one of captions (Karpathy & Li, 2015) and one of bounding boxes(Lin et al., 2014). Both annotation sets are sourced directly from the COCO dataset. First, we notice that of all the weakly supervised methods that use the auxiliary data, AB$_{CLIP}$is the only one that actually improves over CLIP, while both SLIP and SiLC on average decrease in performance relative to CLIP.

**Decoupling annotation and unlabelled data.** Our findings match that of Weers et al. (2023); that invariance-based objectives do not necessarily improve performance, but perhaps instead interpolate between the performance of the invariance objective and the weakly supervised objective. When the invariance objective is "stronger", this leads to improved performance, but in situations like this, where the invariance objective poorly matches the data, it leads to a degradation in performance.

Another approach is to generate annotations by running SimCLR on a well-curated unlabelled dataset like Imagenet, but then running the propagation loss in the t

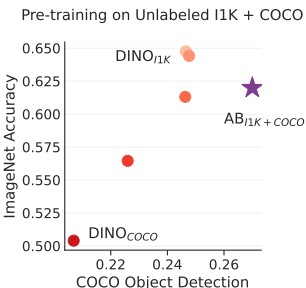

In Figure 3(top), we compare the performance of this mixed approach (we write it as $AB_{I1K+D}$ to running DINO on a mixture of the "clean" Imagenet dataset and the target uncurated dataset $D$. We see that annotation propagation extends beyond the frontier created by running different mixture ratios between the clean and curated data for DINO, learning useful image representations for COCO and EpicKitchens that cannot be acquired by any mixture trained with DINO.

### 5.3 ANALYSIS AND ABLATIONS

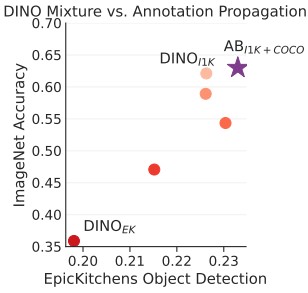

Our results comparing annotation prediction to other self-supervised and weakly-supervised methods indicate that our approach can learn useful semantic features using many different types of base annotations (whether image augmentations, text captions, or bounding boxes). We now more carefully investigate the learning process to determine 1) to what degree is the model actually able to make predictions about the semantic annotations associated with other crops? and 2) what components of the method are important for the obtained performance gains? We run our analysis and ablatory experiments using a smaller data budget of 400M views.

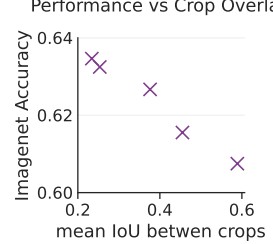

In Figure 5.2 (bottom), we plot the prediction error for the propagation objective throughout the course of training, clustering by how far the target prediction box is in terms of IoU. Notice that prediction errors increase initially in training as the annotation head is first learned, but decreases uniformly through training. We note that the prediction problem appears much more challenging for $AB_{CLIP}$ than for $AB_{SimCLR}$; this is perhaps to be expected since the base loss for SimCLR trains to make the prediction distributions for different bounding boxes as similar as possible.

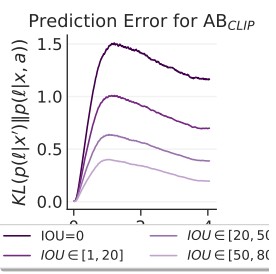

We next investigate how the choice of bounding boxes affects the performance of the algorithm, by sampling source and target bounding boxes that are closer (or further) apart while keeping the marginal distribution over bounding boxes fixed. In Figure 5.2, we see that performance increases steadily as the average IoU between the source and target distributions is decreased; this is perhaps not surprising, as further apart bounding boxes are the hardest to predict, and so training on the most difficult examples offers the strongest learning signal.

Finally, we ablate different components of the method in Table 5.3. As with other self-supervised methods, we find that not using a lagging EMA target network removes all performance gains from the prediction objective (the learning curves entirely mimic that of SimCLR); removing the base loss, which grounds annotation distributions in a semantically meaningful space, also nulls out performance. We also perform an ablation replacing action tokens with null tokens (thereby forcing the model to predict the *average* annotation distribution across nearby images); this reduces the effectiveness of the propagation objective. Perhaps surprisingly, we see that adding image augmentations to either the source or target views actually hurts performance; the general heuristic appears to be that one should select as challenging target images as possible, without introducing any additional stochasticity into the prediction targets (e.g by adding image augmentations or removing action tokens).

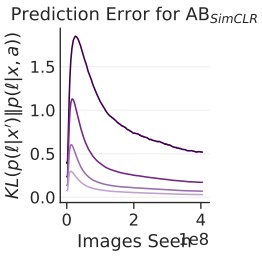

Figure 3: **(top)** Performance of DINO and $AB_{SimCLR}$ training jointly on Imagenet and either COCO or EpicKitchens. **(middle)** Measuring the performance of $AB_{SimCLR}$ as the relative distance between crops to predicts is varied.

| Description | Imagenet Performance |
|---|---|
| Base | $63.0_{-0.0}$ |
| Adding augmentations | $62.5_{-0.5}$ |
| No action tokens | $60.8_{-2.2}$ |
| No propagation loss (SimCLR only) | $59.4_{-3.6}$ |
| No target network | $59.4_{-3.6}$ |
| No grounding loss | $39.0_{-24.0}$ |

Table 4: Ablating different components of $AB_{SimCLR}$

## 6 DISCUSSION

Our paper introduced annotation bootstrapping, a framework for bootstrapping visual representations using unlabeled data that learns by predicting image semantics of the nearby scene. Two qualities make annotation propagation particularly interesting: first, that it cleanly partitions the pre-training process into the specification of image semantics and bootstrapping, allowing us to learn useful details using curated or labeled datasets, while still being able to pre-train on general corpora that do not have the same inductive biases as the curated data. As we saw across a number of datasets, annotation propagation learns useful semantic representations beyond those that are learned from common objectives like pixel prediction, CLIP, or models that learn invariances to crops and augmentations. Our approach is not without limitation; relative to the scale that current CLIP models are being trained on, we were only able to train on relatively small datasets (CC12M – our largest dataset – only has 8 million images) and with relatively small networks (ViT-S), and at limited training durations; some of the conclusions in our paper may weaken at larger scales. Second, while the propagation objective does reduce the dependency on curated datasets and specific inductive biases compared to invariance-based or pixel-predictive approaches, we found the choice of crops to still introduce a form of bias (as we found larger crops to be better). Understanding how we may generalize our approach to more general distortions beyond predicting crops will also be an interesting future direction. Nonetheless, our work takes a step towards understanding how we may pre-train on visual data in a self-sufficient bootstrapped manner using vast swaths of unlabeled data. Already at the larger scales of model pre-training today, we are beginning to see methods bump into the data wall, and we must soon answer the question – how will we improve our models when the data runs out?

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

# A APPENDIX

## A.1 EVALUATION

We use the multi-task decoder-based probe from Beyer et al. (2023) for all of the evaluations in this paper. The probe is defined as a 4-layer transformer decoder with an autoregressive decoding pattern that attends to the final outputs of the Vision Transformer through cross-attention. We choose this architecture so that we can do all of our modelling tasks, whether image recognition or bounding box prediction, or classification of the object in a bounding box using a unified framework; this also represents (albeit to a much smaller scale) how vision transformers are being used in VLM models. We adopt all hyperparameters for training this model from Beyer et al. (2023). Unlike in Beyer et al. (2023), we train probes for individual tasks separately.

When pre-training on Imagenet and CC12M, we probe the model on ImageNet, the Clevr/{Count, Distance} tasks from Zhai et al. (2019b), and then on four tasks used by Beyer et al. (2023): Food101, Oxford IIIT Pets, Resics45, and Sun397.

When pre-training on COCO, we evaluate on small object classification (in which the model is provided the coordinates of a bounding box, and asked to predict the identity of the object within that bounding box), and the corresponding detection task (in which the model must simply identify all bounding boxes corresponding to relevant objects in a scene).

We treat EpicKitchens as a standard dataset of images, considering individual frames independently, and not incorporating the temporal dimension. When pre-training on EpicKitchens, we probe the model also on object classification (predicting the label of an object given its bounding box) and object detection (predicting bounding boxes), which we source from the ViSOR annotation set (Dark-halil et al., 2022). We also probe the model's ability to predict the action a human is taking given one frame of context. This problem is not exactly solvable from one frame of context, but the relative performance differences between methods nonetheless informs the quality of the learned representations.

## A.2 ALTERNATIVE PROBING STRATEGIES

| | METHOD | LINEAR PROBE | ATTENTIVE PROBE | DECODER PROBE | 10-EPOCH FINETUNING |
|---|---|---|---|---|---|
| | SimCLR | 67.0 | 68.7 | 70.0 | 75.0 |
| ImageNet | DINO | **68.5** | 70.0 | 72.2 | 74.2 |
| (No Labels) | MAE | 55.0 | 60.5 | 65.0 | 74.0 |
| | I-JEPA | 58.5 | 61.5 | 64.5 | 72.0 |
| | AB$_{SimCLR}$ | 68.0 | **71.5** | **73.6** | **75.5** |

Table 5: Evaluating models pre-trained on unlabelled Imagenet using different types of probing strategies. We match the evaluation protocol from (Caron et al., 2021), but use a smaller probing length of 10 epochs.

