# OpenReview forum: "Annotation Bootstrapping: Reinforcing Visual Pre-Training using Unlabelled Images"
_ICLR.cc/2025/Conference — Submitted to ICLR 2025_

### Official Review · Reviewer_tkoo · 2024-11-03

**Soundness:** 2
**Presentation:** 2
**Contribution:** 2
**Rating:** 3
**Confidence:** 4

**Summary:**

This paper presents an Annotation Bootstrapping method to learn useful representations from web-crawled unlabeled images. It enforces the self-supervised models to learn to predict the representations of an unseen view based on the given view and the associated transformations from the given view to the unseen view. It achieves better results than previous methods that focus on combing the strength of vision-language contrastive learning (*e.g.*, CLIP)and self-distillation learning (*e.g.*, SimCLR).

**Strengths:**

1. The motivation of utilizing the web-scale unlabeled images is good. It does not limit the method to smaller-scale ImageNet.
2. The improvement is obvious compared with previous methods, such as SLIP and SiLC.

**Weaknesses:**

1. The method is very similar to the practice of I-JEPA [1], which predicts the representations of masked regions based on the learned representations of the visible regions. Slightly differently, this work does not use the masked regions as the optimization target, but uses another cropped view for prediction. It is hard to tell which practice is better, but they are generally very similar.

2. The paper is not very easy to follow. Specifically, it is hard to know what the proposed method exactly is, before reading Equation (5). I am still unsure whether my understanding of the proposed method is correct, by guessing from Equation (5). The first three figures fail to convey enough information of this method. And the caption of the third figure is missing. More importantly, some important details are missing. For example, what is the structure of $g$? How can it accept both the representations and the transformations as inputs?

3. The method is only evaluated under ViT-S. However, as a self-supervised method, it will be more convincing to provide additional results with (at least) ViT-B. The pre-training data has already reached 8M images, therefore, the adopted ViT-S with only 20M parameters may be inadequate to fully unleash the power of self-supervised paradigm. Although the authors have mentioned this point as the limitation, it is still strongly recommend to provide these results.

4. Some typos should be fixed, *e.g.*, there should be a right bracket in L288, and there are two "we" in L245.

[1] Self-Supervised Learning from Images with a Joint-Embedding Predictive Architecture, ICCV 2023

**Questions:**

- Does this method use a pre-training stage to initialize the parameters, or is it simply trained in a self-supervised manner from scratch? From the head figure in page 6, it seems to be a two-stage training framework.
- How well does this method perform when compare with I-JEPA, if trained under the same-scale training data?

---

> ### Author Response · Authors · 2024-11-28
> **Thank you for your review**
>
> Thank you for your thoughtful review! We have updated the paper to include comparisons to I-JEPA in Table 1 and 2; we find that at our data and model scales, annotation bootstrapping outperforms I-JEPA across our domains.  We acknowledge that the current draft of our work has many shortcomings in presentation and clarity, and we greatly appreciate your detailed comments to help us improve the manuscript for the future.Please find responses to your specific questions below:
>
> > How well does this method perform when compare with I-JEPA
>
> We have added a comparison to I-JEPA (using the open-sourced code) for all of the fully unlabelled experiments – the results have been added respectively to Tables 1, 2, and 3 in the updated revision. I-JEPA performs relatively poorly on all of these domains (underperforming SimCLR, DINO, and MAE). This is consistent with the original I-JEPA paper, which found performance to be sensitive to the choice of masking scheme (and consequently, the image distribution) and to smaller model sizes, with large degradations from ViT-H (their primary model) to ViT-B (smallest model they evaluated).
>
> > Does this method use a pre-training stage?
>
> In our experiments, the models are trained from scratch, and all of the pre-training is done simultaneously on the labelled and unlabelled datasets. It would be interesting future work to understand whether this pre-training process can be initialized from a pre-trained model, e.g. a pre-trained CLIP / SigLIP checkpoint.
>
> > What is the structure of g? How can it accept both the representations and the transformations as inputs?
>
> $g$ is a vanilla transformer decoder, identical to a cross-attentive VLM like Flamingo, albeit smaller. This decoder model takes in a sequence of tokens describing the transformation (akin to the “text” in the VLM); at each layer, these tokens cross-attend to the output of the ViT encoder. This allows the model to accept both the image representation (through cross-attention) and transformations (through the inputs).
>
> > The method is only evaluated under ViT-S. It will be more convincing to provide additional results with (at least) ViT-B.
>
> In a future revision, we plan to add an experiment scaling the model size on the CC12M dataset -- we were unable to do so for the discussion period, since we are unfortunately limited by compute.

---

> > ### Comment · Reviewer_tkoo · 2024-11-28
> >
> > Thank the authors for the response. Although the authors conducted quantitative comparisons with JEPA, the overall methodology is still very similar. Besides, I think the current presentation needs to be considerably polished to be published. I would keep my original rating.

---

### Official Review · Reviewer_3wGv · 2024-11-04

**Soundness:** 4
**Presentation:** 2
**Contribution:** 3
**Rating:** 6
**Confidence:** 4

**Summary:**

The paper proposes to bootstrap the annotations under the unlabeled learning and weakly labeled learning setups. The notion of annotations here are universal including human labels such as captions, bounding boxes, or even pseudo-labels (under unlabeled learning setup). The experiment section demonstrates great improvements popular learning algorithms including SimCLR and CLIP.

**Strengths:**

- Novel idea: instead of drawing different views from the same image closer, the paper proposes to use description of the transformations to connect them together.
- Wide range of evaluation datasets

**Weaknesses:**

- Only considers extend one learning algorithm under each setup. Namely, SimClr for unlabeled data learning and CLIP for weakly labeled learning. I'd expect more than one especially for unlabeled data learning since there exists more variants there.
- The abstract is mentioning incorporating captions, bounding boxes, and other annotations, etc. Yet, the experimental section only demonstrate caption-supervised learning algorithms.

**Questions:**

1. Can the authors provide more implementation details? What are the actions (the text describing the transformation) in each dataset (esp Epic Kitchen) look like? What transformations are considered for each dataset/experimental setup?
2. In table 2, the numbers are showing that AB_simclr is worse than simclr in `ImageNet` and `EK Action recog` columns. Can the author elaborate on this?

---

### Official Review · Reviewer_5LF2 · 2024-11-04

**Soundness:** 3
**Presentation:** 2
**Contribution:** 2
**Rating:** 5
**Confidence:** 4

**Summary:**

The paper proposes a framework for pre-training visual features that can incorporate both labelled and unlabelled data. The framework has two main components: the first matches images to its annotations (could be weak labels like image captions or automatic annotations like an augmented view of the image) while given two views of an image, the second makes the model predictive of the annotation distribution of the second view given the first view and the transformation turning the first view to the second view. The approach in the second component is different from other works in the literature which often enforces invariance on the prediction on the two views.

The papers shows better results than competitors in both self-supervised learning and text-image aligning tasks.

**Strengths:**

The framework discussed in the paper provides a view that unifies certain self-supervised methods such as DINO and Swav and image-text alignment methods such as Naeem et al.. This view is interesting. In the fully unlabelled setting, the main difference between the proposed framework and DINO or Swav is the consideration of the transformation, or actions, that transform one view to another. This seems to implicitly enforce equivariance instead of invariance on the model's prediction on the two views, and makes the framework more robust when cropping invariance does not apply.

The paper shows that the proposed method improves upon baselines in multiple benchmarks.

**Weaknesses:**

There are several issues with the paper's presentation:
  - The presentation of the method is overly complex with unnecessary and under-clarified equations such as Eq. 1. In particular, what is the space that the right term is optimized on? What are l and x in the left term? Are they variables or given fixed image and annotation? The inclusion of this equation seems unnecessary.
    The whole framework is very similar to the common framework used for semi-supervised learning [1, 2] which often consists of a superivsed loss computed on labelled data and an unsupervised loss to enforce invariance or equivariance [3] on the model's prediction. In this case, the L_annotation loss corresponds to the supervised loss and the L_propagation loss amounts to implicitly enforcing equivariance with respect to the transformation "a" on the model's prediction. From this point of view, the proposed framework is very similar to DINO or Swav consider multi crops and enforce consistency between global crops (L_annotation) and between global crops and local crops (L_boostrapping without considering the transformation). The projection of image features to the annotation space is also similar to the projection to the space of prototypes in DINO/Swav. It is also very similar to Naeem et al. in the case of text-image alignment. The paper should remove unnecessarily heavy maths  and make the connnections to DINO/Swav and Naeem et al. clearer.
  - The paper contains many typos such as missing citation "cite" (L.92, 97), wrong refs (Figure 5.2 in L432, 455), inconsistent font size in Fig. 3, short, uninformative captions for Tables 2-3

[1] S4L: Self-Supervised Semi-Supervised Learning. Zhai et al., 2019

[2] Revisiting Consistency Regularization for Semi-Supervised Learning. Fan et al., 2022

[3] A Survey on Deep Semi-Supervised Learning. Yang et al., 2022

**Questions:**

Other comments and suggestions:
  - L.13-15: This is not correct. Many existing methods work learn well from web datasets or even satellite images (DINOv2, Tolan et al. [1], Vo et al. [2], AIM [3]).
  - L.164: p(x,l) seems to be the joint image-annotation distribution, not the annotation distribution
  - How exactly COCO captions and bounding boxes are used as annotation in Table 3? Which dataset/annotation is used in contrastive annotation loss and predictive propagation loss?
  - L431. How to generate annotation with SimCLR, what are the annotations?
  - claims in L.432-439 are not supported by the figure without results on DINO/1K+COCO
  - L 464-471. If smaller IoU leads to better performance, why don't we train on non-overlapping crops?
  - The selection of l_i from the annotation batch in the annotation boostrapping update in Algo 1 seems arbitrary since the images are different in the two updates. It seems that a any projection of the image or image/action embeddings into the annotation embedding space would work here. In this spirit, this projection is very similar to the projection to the prototype space in DINO. The main difference is the addition of the action tokens in the loss equation. What is the performance if learnt prototypes are used as in DINO instead of such arbitrary projection to the annotation space?

[1] Very high resolution canopy height maps from RGB imagery using self-supervised vision transformer and convolutional decoder trained on aerial lidar. Tolan et al., 2024

[2] Automatic Data Curation for Self-Supervised Learning: A Clustering-Based Approach. Vo et al., 2024

[3] Scalable Pre-training of Large Autoregressive Image Models. El-Nouby et al., 2024

---

> ### Author Response · Authors · 2024-11-28
> **Thank you for your review**
>
> Thank you for your thoughtful review!  We acknowledge that the current draft of our work has many shortcomings in presentation and clarity, and we greatly appreciate your detailed comments to help us improve the manuscript for the future. Please find detailed responses to your questions below:
>
> > L 464-471. If smaller IoU leads to better performance, why don't we train on non-overlapping crops?
>
> We do! In our main experiments, we use the standard random resized crop RRC(8, 100) to sample crops (which w/ some probability, results in non-overlapping crops). The experiment in Figure 3 was a controlled experiment to determine the impact of the overlap between different crops
>
> > L.13-15: This is not correct. Many existing methods work learn well from web datasets or even satellite images (DINOv2, Tolan et al. [1], Vo et al. [2], AIM [3]).
>
> We apologize for the over-statement. Our intention was simply that careful curation of these raw web-scale datasets  is an important ingredient of many of these efforts, e.g. exemplar-based automatic data curation in DINOv2, and ImageNet mixture sampling in AIM.
>
> > From this point of view, the proposed framework is very similar to DINO or SwaV… (L_boostrapping without considering the transformation)
>
> The functional form of our update indeed looks similar to DINO/SWAV, but the choice of “prediction space” leads to significantly different learning styles. In DINO, the prediction space is recursive, predicting the features that themselves were used to optimize the prediction objective. In contrast, for annotation bootstrapping, the prediction space is not recursive; the model is requested to predict the “contrastive features” (coming from L_annotation), not the “bootstrapping features” (coming from L_propagation). The former encourages the encoder to learn features that can be easily predicted, leading to features that are shared and inferrable across “local” crops of an image. The bootstrapping objective makes no such inclination towards learning more (or less) predictable features, but rather defers the choice of what features to predict to the annotation objective. We hope to make the distinctions between our method and DINO / SWAV more explicit in a future revision of the methods section.
>
>
> > How exactly COCO captions and bounding boxes are used as annotations in Table 3?
>
> Bounding boxes and captions are tokenized similar to standard VLM training pipelines. Bounding boxes are tokenized into the following format “<y0=0.1> <x0=0.0> <y1=0.5> <x1=0.3> Object Name”; Caption texts are directly tokenized. In this experiment, both the unlabelled and labelled training is done using COCO.
>
> > What is the performance if learnt prototypes are used as in DINO instead of such arbitrary projection to the annotation space?
>
> This is an interesting experiment to run, and one we will add to the paper! We note though that the projection to the annotation space is not arbitrary; in fact, this projection means that both the annotation loss and bootstrapping loss are making predictions about the same underlying distribution (the annotation distribution associated with an image), in principle offering a canonical way to balance the two losses. It is not obvious whether this will hold when using learned exemplars as in DINO.

---

> > ### Comment · Reviewer_5LF2 · 2024-12-03
> > **Response to authors**
> >
> > I would like to thank the authors for their answers to many (not all) of my questions. I believe that the paper is interesting if it can be rewritten in a clearer way. In particular, unnecessary maths should be removed, Algo 1 should be better explained (I still think that the choice of the second update is quite arbitrary) and the connection to I-JEPA and DINO/Swav should be clarified. With only the authors' responses and without an update to the manuscript, there is not enough evidence to increase my rating. Therefore, I will keep my original score.

---

### Official Review · Reviewer_YpYr · 2024-11-09

**Soundness:** 3
**Presentation:** 1
**Contribution:** 3
**Rating:** 3
**Confidence:** 3

**Summary:**

- The paper presents annotation bootstrapping, a learning approach which trains a model to predict semantic “annotations” and “bootstrapping” by predicting the annotations for a different crop given a crop
- The method supports SSL training, WSL, and semi supervised
- The paper shows AB with two objectives, SimCLR and CLIP
- The paper shows results on various tasks by training on image and video datasets

**Strengths:**

- The paper starts with a good intuition, and proposes to use annotations to apply an SSL loss where one crop predicts the annotations for another crop
- The results seem strong (albeit in a very uncommon settings)
- The approach supports SSL, image-text, semi-labeled data, etc. extending its applicability

**Weaknesses:**

- The paper is quite hard to understand. Figure 2 works explain things at a high level, but doesn’t explain the exact approach. Section 3 also talks about mathematical equations but doesn’t discuss practical details. Finally Algorithm 1 doesn’t help clarify details either. The annotation l is a function of the input x, so if x is transformed, l should be different. But since it isn’t defined as a function, it becomes an overloaded term. Input x has an annotation l, but also we try to find l given a transformation a on x. In Algorithm 1 in annotation bootstrapping, is $g_\theta(x, a, l)$ the simCLR loss applied to the transformed image? SimCLR itself applies transforms so it’s not clear what exactly is happening practically to me. What does “ copy (only) the annotations from the labelled batch” mean in L281? While bootstrapping are we using both labelled and unlabeled images? What does “copy” the annotations mean – we anyway create new crops for an image during bootstrapping so we don’t have real annotations for these crops if I understand things correctly.
- The evaluations are performed with a single setup of using a decoder-based probe which cross attends to the ViT outputs. While this is an interesting setup, it is not commonplace. This makes the results hard to compare vs. “regular” setups like linear probing, finetuning for the tasks, etc. It is not clear if AB performs better in this setting vs. other settings. Especially for a new method like AB, it is very important to have evaluation results in more common settings like linear probes and finetuning, at least on ImageNet, and then making comparisons
  - MAE only works well when finetuned, so the current setup puts it at a disadvantage anyway
- There are lots of typos, mistakes, etc. which again make the paper hard to read, e.g. Table 1 says AB_{CLIP} on ImageNet (No Labels) whereas the discussion refers to this as AP_{SimCLR} – it has to not be CLIP since there’s no labels I assume. Table 2 bolds MAE for some columns even though the results are not best. It is not clear whether the numbers are incorrect or if there is some other mistake? There is no Figure 5.2 in the paper (L432). I see Figure 3, but I don’t see anything in the plots referring to $AP_SimCLR(I1K)$
- How is the video training performed? There are no details about the approach around EK.
- There should be comparisons to JEPA style architectures like I-JEPA, especially since even there the model is trained to produce representations (not annotations) for one crop from another crop

**Questions:**

Overall, the paper doesn't feel like it was ready for submission. To even consider recommending acceptance, I would need the following details:
- Please clarify fundamental details about the approach
- Please show evaluations using more standardized setups

Other than that please look at the weaknesses

---

> ### Author Response · Authors · 2024-11-28
> **Thank you for your Review**
>
> Thank you for your thoughtful review. We have updated our submission to include 1) results for I-JEPA and 2) traditional probing setups, including linear probing and full finetuning on Imagenet (Table 5), . We acknowledge that the current draft of our work has many shortcomings in presentation and clarity, and we greatly appreciate your detailed comments to help us improve the manuscript for the future. Please find responses to specific questions in your review below:
>
> > There should be comparisons to JEPA style architectures like I-JEPA
>
> We have added a comparison to I-JEPA (using the open-sourced code) for all of the fully unlabelled experiments – the results have been added respectively to the updated revision. I-JEPA performs relatively poorly on many of these domains (underperforming SimCLR, DINO, and MAE). This is consistent with the original I-JEPA paper, which found performance to be highly sensitive to the choice of masking scheme (and consequently, the image distribution) and to the model size, with large degradations from ViT-H (their primary model) to ViT-B (smallest model they evaluated).
>
> > Please show evaluations using more standardized setups
>
> We have updated the submission with Table 5, which measures Imagenet linear probing accuracy and full fine-tuning accuracy on the models pre-trained on Imagenet. Annotation bootstrapping remains the strongest method under full finetuning, although DINO performs slightly better under a linear probe. We note that linear probing disproportionately downweights 1) pre-training methods that do not pool encoder outputs in pre-training like MAE and I-JEPA, and 2) methods that pre-train on datasets visually distinct from Imagenet (especially COCO and Epic Kitchens).
>
> > What does “ copy (only) the annotations from the labelled batch” mean in L281? While bootstrapping are we using both labelled and unlabeled images?
>
> The bootstrapping objectives uses only the unlabelled images, but still needs a set of annotations to generate a target distribution (i.e. to predict $p(\ell | s, a)$, we need to plug in some $\ell$). Since the unlabelled and labelled pre-training happens concurrently, the easiest way to implement this is to “copy” the list of annotations from the labelled batch to use for the unlabelled update. These two lists do not need to be the same however: we can use a memory bank of previously seen annotations or we can use a fixed set of annotations for the bootstrapping update.
>
> > How is the video training performed? There are no details about the approach around EK.
>
> We treat EpicKitchens as a dataset of individual frames, to match the setup with the other domains in the paper. We have updated the paper to include this detail.

---

### Meta-Review · Area_Chair_dgzS · 2024-12-19

**Metareview:**

As the reviewers noted, the paper presents good intuition, coverage, and practicality. However, multiple reviewers pointed out presentation issues (e.g., hard-to-understand, complex, and unclear presentation) that need improvement. In addition, while the authors provided experimental comparisons with I-JEPA during the rebuttal period in response to reviewers' concerns, they failed to clearly demonstrate and explain the advantages of their proposed method and architecture over I-JEPA in a convincing manner. Therefore, I recommend the rejection of this paper.

**Additional Comments On Reviewer Discussion:**

- `Reviewer YpYr`, `Reviewer 5LF2`, and `Reviewer tkoo` raised concerns about the paper's presentation quality. Given that improving writing quality during the rebuttal period is difficult, we hope to see these aspects addressed in the next submission.
- `Reviewer YpYr` and `Reviewer tkoo` pointed out the similarities between the proposed methods and I-JEPA. They also noted the lack of comparisons. While the authors attempted to defend their work with experimental results, they failed to provide sufficient justification for what and why the proposed method is superior to I-JEPA.

---

### Decision · Program_Chairs · 2025-01-22

Reject